# Insights into the Control of Drug Release from Complex Immediate Release Formulations

**DOI:** 10.3390/pharmaceutics13070933

**Published:** 2021-06-23

**Authors:** Runqiao Dong, James C. DiNunzio, Brian P. Regler, Walter Wasylaschuk, Adam Socia, J. Axel Zeitler

**Affiliations:** 1Department of Chemical Engineering and Biotechnology, University of Cambridge, Philippa Fawcett Drive, Cambridge CB3 0AS, UK; rd489@cam.ac.uk; 2Pharmaceutical Sciences, Merck, Rahway, NJ 07065, USA; james.dinunzio@merck.com (J.C.D.); brian.regler@boehringer-ingelheim.com (B.P.R.); walter_wasylaschuk@merck.com (W.W.); adam_socia@merck.com (A.S.)

**Keywords:** pharmaceutical tablet, terahertz, terahertz pulsed imaging, MCC, wetting, NaCl, disintegration, dissolution, porosity

## Abstract

The kinetics of water transport into tablets, and how it can be controlled by the formulation as well as the tablet microstructure, are of central importance in order to design and control the dissolution and drug release process, especially for immediate release tablets. This research employed terahertz pulsed imaging to measure the process of water penetrating through tablets using a flow cell. Tablets were prepared over a range of porosity between 10% to 20%. The formulations consist of two drugs (MK-8408: ruzasvir as a spray dried intermediate, and MK-3682: uprifosbuvir as a crystalline drug substance) and NaCl (0% to 20%) at varying levels of concentrations as well as other excipients. A power-law model is found to fit the liquid penetration exceptionally well (average $R^{2}>0.995$). For each formulation, the rate of water penetration, extent of swelling and the USP dissolution rate were compared. A factorial analysis then revealed that the tablet porosity was the dominating factor for both liquid penetration and dissolution. NaCl more significantly influenced liquid penetration due to osmotic driving force as well as gelling suppression, but there appears to be little difference when NaCl loading in the formulation increases from 5% to 10%. The level of spray dried intermediate was observed to further limit the release of API in dissolution.

## 1. Introduction

Immediate release tablets are designed to rapidly disintegrate and release drug substance after administration. Disintegration is a complex process that involves a combination of dissolution medium wicking into the porous tablet, particle swelling, strain recovery and rapid dissolution [1]. These processes are in turn affected by the interplay of formulation and processing conditions which determine the physical properties and microstructure of the tablets [2]. Therefore, having a clear understanding by breaking down each attribute and their contributions is important for drug product and process development.

However, the pharmacopeial disintegration test results in a single value of disintegration time for a batch of tablets under test. It is therefore of relatively limited use as a stand-alone tool to gain quantitative or mechanistic insight into the process.

As a result, the disintegration process is in practice mostly studied indirectly through measuring the release and subsequent dissolution of the API in dissolution testing. The dissolution process has attracted widespread interest in pharmaceutical analysis with examples of such studies including pharmacopeial dissolution testing using UV-Vis/HPLC analysis [3,4], MRI methods [5,6,7], thermo gravimetric analysis [8,9], infrared spectroscopy and infrared imaging [10,11,12] as well as UV imaging [13] to name a few. These techniques can be used to aid the understanding of the release kinetics of the API, investigate relatively slow (timescales of tens of seconds) tablet swelling processes as well as to provide insight into how tablet formulation, geometries and dissolution kinetics affect API release during formulation development.

Terahertz time-domain spectroscopy (THz-TDS) and terahertz pulsed imaging (TPI) have received considerable research interest in the pharmaceutical industry over the past two decades. The methods are safe, non-destructive and fast measuring techniques, whilst exhibiting excellent potential for exploring the inter-molecular structure and dynamics of organic molecular solids as well as being able to probe the microstructure of solid dosage forms [14].

In this study, TPI is coupled with a novel design of a flow cell. In this cell the tablet is held horizontally in a fixed position and is exposed to a laminar flow of dissolution medium at constant velocity at its lower face. Terahertz time-domain reflection measurements are performed through the dry upper face of the tablet [15]. The use of the flow cell setup makes it possible to separate the wicking process of dissolution medium into the tablet from shear erosion under turbulent flow conditions. It is well-known that such mechanical shear can accelerate tablet disintegration during pharmacopeial tests. Unlike the single value of a representative disintegration time from the pharmacopeial test, the TPI setup captures real-time data of the entire process of water penetrating through a tablet. Simultaneously, the concomitant swelling behaviour of the tablet matrix in liquid propagation direction can be extracted from the terahertz waveforms.

Previously, the method has been used to measure liquid imbibition within tablets of pure microcrystalline cellulose (MCC) as well as tablet formulations made from MCC containing disintegrant [16]. Subsequent work highlighted the potential of this method to study much faster liquid transport processes (on sub-second timescales) at the example of highly porous tablets made from functionalised calcium carbonate (FCC) [17]. In addition, the time-dependent swelling behaviour of pure MCC tablets has been investigated in more detail [18].

For the present study, considerably more complex formulations were investigated: a fixed-dose combination (FDC) product containing solid dispersion intermediate (SDI) and crystalline API was designed with varying levels of SDI and sodium chloride (NaCl), and tablets were compacted over a range of porosities. As a key quality attribute for any immediate release tablet, porosity determines how fast water can travel into the tablet matrix by capillary action [19]. It can be directly controlled by the particle size of the formulation and the processing parameters during powder compaction. However, the compaction behaviour is highly specific to the formulation and the overall dissolution rate of a tablet is related to both porosity and formulation. Notably, amorphous dispersion products present unique challenges for rapid disintegration due to the high fraction of hydrophilic polymer used in preparation of the material [20]. It is believed that the reduced disintegration and dissolution of tablets, particularly at higher levels of SDI, is the result of gel formation of sufficient mechanical integrity to retain the monolithic character of the tablet. Better understanding of the mechanisms responsible for dissolution behaviour of SDI containing tablets could potentially lead to direct improvement on tablet performance.

On the other hand, NaCl is commonly used as a pore former [21] as well as an osmogen [22]. Furthermore, recent studies have shown that adding inorganic salts, such as NaCl, can accelerate dissolution by suppressing polymer gelling and thus facilitating drug transport processes [23,24,25]. In terms of the latter process the magnitude of the effect was found to depend on the particle size of the salt, pH as well as the types of polymer used in the formulation [26,27,28]. However, extra care needs to be taken when deciding on the amount of NaCl to include in the formulation as (i) the solubility of the API may be compromised, especially if APIs are supplied as a hydrochloride or sodium salt form; (ii) the effectiveness of swelling disintegrants might be suppressed due to ionic effect; (iii) the stability of tablets during storage might get compromised due to the enhanced moisture sorption with NaCl in the formulation and (iv) it is desirable to minimise the total amount of NaCl due to its potential impact on high blood pressure and other physiological effects in the patient [29,30].

In this paper, we aim to establish the impact of the changes in formulation as well as porosity on the liquid transport processes and explore their interplay for the performance and critical quality attributes in complex immediate release tablets. TPI data are also to be compared to results from USP disintegration and dissolution tests.

## 2. Materials and Methods

### 2.1. Materials

All tablets used in this study were made by Merck Sharp & Dohme Corp, a subsidiary of Merck & Co, Inc, Kenilworth, NJ, USA (MSD), using the formulations shown in Table 1. They were designed specifically so that the effect of different levels of SDI and sodium chloride on drug release can be investigated. The SDI contained a solid dispersion of ruzasvir, HPMC and vitamin E TPGS. Note that the concentration of SDI is complementary to MCC.

Tablets that were used for the liquid transport study were of 10 mm diameter and with thickness controlled in the range of 1.5 mm–2.0 mm. For each formulation, two levels of porosity (10 ± 1% and 16 ± 1%) were prepared by varying the compaction forces in order to investigate the influence of porosity on the water ingress (Table 2). At each compaction level, the stress applied was kept constant for all formulations. The low porosity batches were designated the identifier L, i.e., F01L for formulation 1 compacted to low relative porosity, and the corresponding high porosity batch is referred to as F01H. For each batch, three tablets were used for the measurements and the average from the three measurements were used for the subsequent analysis.

The nominal porosity for individual tablets was calculated by using Equation (Equation 1):(1)$$f=1-\rho_{r}=1-\frac{w}{V\rho_{t}}=1-\frac{4w}{\pi d^{2}h\rho_{t}}$$
where *w* is the mass of tablets, measured by a balance with precision to 0.1 mg. *h* is the thickness of tablets, *d* is the diameter of tablets. Both properties were measured using a digital micrometer. $\rho_{t}$ is the true density, which was estimated from the direct measurement of individual component true densities.

### 2.2. Methods

#### 2.2.1. Terahertz Pulsed Imaging (TPI) Experiments

A commercial time-domain terahertz system (TeraPulse 4000, TeraView Ltd., Cambridge, UK) equipped with a fibre-based flexible refection probe was used to acquire the terahertz time-domain waveforms. The reflection probe was equipped with a silicon lens of focal length of 18 mm. The beam waist at the focus was ≈600 μm, and the focal point was adjusted in *z*-direction, i.e., the propagation direction of the terahertz pulse, to the centre of the tablets so that a good signal of both back and front faces can be recorded from a single terahertz waveform.

The system was configured to operate at an acquisition rate of 15 Hz. Given that the liquid transport processes were slower than in previous experiments and did not complete at sub-second timescales a scan co-average of 15 was used to improve the signal-to-noise ratio whilst reducing the amount of data to be processed from the experiments. As a result, one reflection waveform per second was generated. A schematic of the experimental setup is shown in Figure A1 in the Appendix A and further detail was reported previously [15]. The fundamental principle of data acquisition and data processing of this method has been explained in detail in previous studies [16,17].

The surface area of the tablet exposed to the liquid was 48.4 mm^2^. The flow of dissolution medium in the cell was controlled by valves at the inlet and the two outlets. The flow cell was sealed with a polyethylene window (PE), which is transparent to terahertz radiation and results in little attenuation to the signal from desired interfaces [17]. The temperature of water used was 20 °C and the flow rate was kept constant at 13 mL min${}^{-1}$.

For each tablet, the data acquisition was initiated just before water touched the tablet from below and was stopped only after disintegration was completed. An example of the resulting TPI data for the measurements captured for one tablet is shown in Figure 1a. In order to investigate the dynamics of water ingress quantitatively, the water front displacement was plotted against time as demonstrated in Figure 1b. Whether or not the water front can be tracked over the entire thickness of the tablet depends on the relative contrast from the reflection originating from the water front. Initially, there is a well defined separation between hydrated and dry tablet which results in a clear peak given the difference in refractive indices between the two structures. The more diffuse the interface between the water phase and the dry porous tablet matrix becomes during the transport process, i.e., a water concentration gradient forms, the reflection peaks looses contrast and becomes more difficult to track.

#### 2.2.2. Complementary Testing

USP dissolution and disintegration tests were performed on tablets made from the same formulations. Dissolution testing was conducted using USP Apparatus I containing 900 mL of 0.3% (*w*/*v*) SDS in 20 mM, pH 6.8 sodium phosphate buffer operated at 100 rpm. Disintegration testing of tablets was performed using USP reciprocating cylinder apparatus containing purified water. Tablets used for this testing were larger modified oval tablets intended representing the intended dosage form size. The smaller tablets used for terahertz testing and larger tablets for dissolution and disintegration were considered to have similar porosity.

As mentioned previously, the TPI experiments exclude any contributions from mechanical agitation and shear exerted on the tablets that originate from the turbulences in the dissolution vessel used in the USP tests.

## 3. Results and Discussion

### 3.1. Liquid Transport Data

The liquid penetration profiles of all batches were plotted for each study together with the dissolution profiles of the corresponding formulations to allow for comparison (Figure 2). The extent of dissolution, as determined by USP dissolution testing, is expressed in terms of a faction of drug dissolved over the range of 0 to 1 where 1 corresponds to complete drug release and dissolution.

#### 3.1.1. Study 1: Investigation of Different Concentrations of SDI

The results show that at the higher porosity level of 16%, neither the amount of SDI nor the relatively small porosity variation between tablets appears to affect the rate of water ingress into the tablets very much (Figure 2a).

When the porosity was reduced to about 10% the overall liquid transport rate decreases as expected and it is the relative concentration of SDI in the formulations that governs the rate of ingress of water into the tablets (Figure 2b). Higher relative concentrations of the swelling MCC particles result in faster liquid transport. The liquid transport measured for sample F02H was the slowest. This could be due its porosity being 2% lower compared to the other high porosity batches, hence slowing down mass transport.

On the other hand, the results from the USP dissolution tests (Figure 3c,d) showed that the gradients of dissolution profiles are always negatively correlated to SDI loading, indicating an apparent sensitivity of dissolution rate to SDI, even for the high porosity batches.

The comparison of relative impact from SDI and porosity will be discussed in more detail in Section 3.3.

Overall, compaction at high pressures was found to slow down water ingress as well as the dissolution rate, and porosity can be considered the main rate-limiting factor for drug release and dissolution in such cases. Meanwhile, the dissolution rate was found to be more strongly correlated to the SDI content than water ingress rate, especially at high porosity.

#### 3.1.2. Study 2: Investigation of Different Concentrations of NaCl

The four formulations in Study 2 can be divided into two sub-groups according to their SDI levels to make comparison easier: F02 and F05 with the low concentration of SDI; and F06 and F07 with relatively high concentration of SDI.

From the data shown in Figure 3a,b, it can be established that liquid penetration is considerably slowed down by reducing the amount of NaCl in the formulation. However, there appears to be little difference when concentration is increased from 5% to 10%, which indicates that at 5% of NaCl the process is no longer limited by NaCl concentration in the formulation.

The data from the dissolution tests are in agreement (Figure 3c,d), but in addition, the influence from SDI concentration on dissolution rate can be also clearly observed. When comparing the two sub-groups, it can be noticed that the relative impact from varying SDI concentration is more apparent at low porosity.

### 3.2. Swelling

In addition to analysing the water ingress into the tablets the data shown in Figure 1a can be used to investigate the swelling process in liquid propagation direction by tracking the peak maximum representing the position of air/tablet interface as a function of time.

As shown in Figure 4a,b for Study 1, the extent of swelling appears to be lower at a higher SDI loading. While the HPMC in the SDI forms a swelling gel structure upon hydration that acts as a diffusion barrier, inhibiting further hydration of tablet, it needs to be stressed that a relatively large fraction of MCC is present in the formulation, which is complementary in quantity to the amount of SDI (Table 1). MCC swells much more quickly than HPMC upon absorbing water and does not form a mass transport limiting gel in the same way HPMC does. It will thus promote the hydration throughout the dosage form. Given how the timescales of the swelling process (for example see F05 in Study 2, Figure 4c,d) match the liquid penetration process (Figure 3), it is clear that the swelling is limited by the extent of liquid penetration. It is therefore interesting to compare the amount of swelling at a given penetration depth, which will be discussed in Section 3.3.

In study 2, there is some evidence that NaCl facilitates swelling in addition to the effect of SDI (Figure 4). It is worth highlighting that overall we find qualitative agreement between the shape of the swelling curves and the USP dissolution profiles (Figure 3c,d), corroborating that volume expansion due to swelling and strain recovery is a major mechanism that contributes to the disintegration and dissolution processes. This in turn explains why the impact from SDI concentration on dissolution is much larger compared to just liquid penetration.

### 3.3. Quantification

#### 3.3.1. Liquid Penetration

To investigate the trends in liquid transport quantitatively and facilitate further analysis, the averaged liquid transport data from all runs of the same batch were fitted using a power law equation:(2)$$y=kt^{m}$$
where *y* is the extent of water penetration, *t* is the time and k,m are the parameters for power-law fitting, The average $R^{2}$ for all fits was 0.995.

Either *k* or *m* alone cannot directly indicate how fast the liquid penetration process is. In order to facilitate the comparison, an apparent rate of liquid penetration *P* was defined by finding the time taken for water to penetrate the first mm of dry tablet thickness: $t_{1\;\mathrm{mm}}=k^{-m^{-1}}$, thus:(3)$$P=\frac{1\;\mathrm{mm}}{t_{1\;\mathrm{mm}}}=k^{m^{-1}}$$

A distance of 1 mm was chosen since the total thickness of the tablets was around 2 mm, and by selecting half of the thickness a good estimate of the rate can be obtained before symmetry is crossed. The values of *P* and the fitting parameters *m* were plotted against porosity and MCC concentration (Figure 5). The factor *m* provides information on the water transport mechanism. The error bars show the highest and lowest value among samples of the same batch. Note that the high porosity batches exhibit larger uncertainty from tablet to tablet, which could be explained by a less confined flow path due to the more porous microstructure of these tablets.

Overall, we found linear correlation between the water ingress transport rate and power law exponent and porosity as well as SDI concentration (Figure 5). The liquid transport rate, *P*, increases in a linear fashion with increasing porosity, showing the potential of regulating the rate of water ingress in a predictable manner. This is in excellent agreement with previously observed trends in simple immediate release tablets based on a MCC formulation [1].

At the same time we observe a linear decrease of *m* with porosity, which is also consistent with the results from a previous study by [16]. The higher the porosity the more the liquid transport characteristics resemble Darcy flow (m=0.5), while the less porous samples share characteristics which more resemble swelling-controlled case II transport (m=1). The data points depicted by yellow squares and highlighted by a dashed red circle in Figure 5, correspond to sample F05. This sample does not contain any NaCl and its values deviate very clearly from the linear trend. This observation emphasises how significant the effect of NaCl is on the liquid ingress into the tablet matrix. The upward shift of *m* in sample F05 compared to the other samples, also suggests that without NaCl present in the formulation, which is known to suppress the hydration of polymers [31], the flow characteristics more closely resemble the swelling-controlled case II transport.

#### 3.3.2. Disintegration and Dissolution Test Results

The USP test data were analysed using a similar methodology and a dissolution rate constant was obtained from fitting a linear function to the data points before the dissolution has reached a plateau in the release profile. The disintegration test only yields a single data point, the disintegration time, and hence the tablet mass was divided by the time value to yield a disintegration rate in units of mg·min^−1^ for each tablet. The value of an average of 3 tablets was quoted here for each batch. We found excellent linear correlation between the dissolution and disintegration data (Figure 6a). Therefore, dissolution data will be quoted alone to represent USP test results for the subsequent analysis. To account for variation in tablet geometry, the dissolution rate ($\min^{-1}$) was further normalised by the surface-area-to-volume ratio (${\mathrm{mm}}^{-1}$) for comparison with liquid transport data.

Like for the liquid transport data, we also observe a linear correlation between the dissolution rate and porosity (Figure 6b) as well as SDI loading (Figure 6c). However, dissolution rate appears to be strongly influenced by the concentration of SDI. Even in the plot of rate against porosity (Figure 6b), dissolution rate display clear trend with SDI concentration, and this feature is not observed for liquid transport (Figure 5a).

Figure 6d shows some degree of linearity between rate of liquid penetration and dissolution processes. However, as SDI concentration increases, liquid penetration rate would over predict the dissolution rate.

#### 3.3.3. Swelling

The swelling curves shown in Figure 4 mostly follow an S-shape. When water enters the tablets from the bottom, the wetted volume of the tablet that is swelling pushes against the weight of the tablet as well as the pressure exerted from the washers around the rim of the tablet that fix it into place on top within the flow cell sample holder. This constraint results in the plateau that is observed with regards to the extent of swelling after the initial expansion period. However, when the remaining dry volume at the top of the tablet becomes sufficiently thin as the liquid continuous to penetrate into the dry porous matrix, the force due to the swelling process is sufficient to overcome the force exerted on the tablet centre from the retaining washers at the rim and the top surface, the air/tablet interface detected by terahertz radiation, can continue to move upwards. This behaviour is specific to the setup in our experiment, but the initial swelling curve is still believed to be a good representation of the early swelling response of the tablet formulation.

Therefore, the inflection points were found by fitting a polynomial function to the swelling curves, and an apparent swelling rate *S* was defined as the amount of volume expansion ($E_{\mathrm{inflection}}$ in mm) divided by the time taken ($t_{\mathrm{inflection}}$ in min) to reach the point of inflection:(4)$$S=\frac{E_{\mathrm{inflection}}}{t_{\mathrm{inflection}}}$$

The extent of swelling at a fixed penetration depth was also determined in analogy to the methodology discussed for the apparent liquid penetration rate *P* (Equation (Equation 3)). The resultant parameter was $E_{1}$, the extent of swelling at 1 mm penetration depth.

An illustration of how *S* and $E_{1}$ change against porosity and SDI concentration is shown in Figure 7. Both *S* and $E_{1}$ exhibit negative correlation with SDI concentration. *S* appears to show greater linearity with porosity than $E_{1}$ since the rate of swelling is dependent on rate of liquid penetration which is strongly correlated to porosity, while for $E_{1}$, that dependency is absent.

### 3.4. Factorial Analysis

Based on the results, we found evidence that the kinetics of the liquid ingress into the tablets, swelling as well as dissolution clearly depend on more than a single ingredient within the formulation or any single factor that is characteristic of the dosage form microstructure. Therefore, a factorial analysis was performed to compare the magnitude of impact from the main factors we identified, namely tablet porosity ($X_{1}$), the concentration of SDI ($X_{2}$) and the concentration of NaCl ($X_{3}$) in the tablet formulation. A simple linear model was fitted using multiple regression analysis as follow:(5)$$Y=a_{0}+a_{1}X_{1}+a_{2}X_{2}+a_{3}X_{3}$$
where $a_{0}$ is the intercept and $a_{1},a_{2},a_{3}$ are the regression coefficients. *Y* is the dependent variable and is set to represent different aspects of the performance of a tablet in dissolution and liquid penetration, i.e., dissolution rate (*D*), apparent liquid penetration rate (*P*), power index of liquid penetration (*m*), apparent swelling rate (*S*) and the extent of swelling at 1 mm liquid penetration ($E_{1}$). In order to be able to compare relative effects the absolute values of different *Y*s were all scaled to unit variance, e.g., $D/\sigma_{D}$, where $\sigma_{D}$ is the standard deviation of *D*, so that the resulting coefficients can be directly used to compare the magnitude of impact across different *Y*.

All the regression coefficients that resulted from this analysis are listed in Table 3 along with the determination coefficients ($R^{2}$) and the *p*-values from the *t*-test of each coefficient and *p*-values from the ANOVA *F*-test for each *Y*. A significance level of p>0.05 was applied.

Based on the $R^{2}$ values, both dissolution rate *D* and liquid penetration rate *P* strongly correlate with the independent variables, among which porosity seems to have a dominating impact with $a_{1}=30.24$ and 26.52 respectively. The liquid penetration rate appears to be relatively insensitive to the SDI concentration based on the result of a non-significant $a_{2}$ value. However, the results from the analysis indicate that the liquid penetration rate is more affected by NaCl concentration than dissolution rate (13.51>9.207), which means that the osmotic effect provided by NaCl and the suppression of gelling of SDI by NaCl [26] plays a bigger role for liquid penetration.

A low loading (10wt%) of SDI can already form enough gel to slow down bulk water ingress, hence it mainly relies on the NaCl concentration in the formulation to improve the liquid penetration rate. But for the dissolution process, it also matters how fast API can be released from the gel in addition to water ingress. A higher SDI loading means that a higher fraction of API will be released slowly through the gel even after the full hydration of the tablet matrix, and, therefore a significant negative value of $a_{2}=-6.220$ is seen.

It is important to note that in the TPI setup, very little shear erosion is applied to the wetted tablet whilst in a USP test such shear in the turbulent conditions of the USP testing apparatus can help erode any emerging gel layer on the outside of the dosage form and facilitate disintegration.

For the parameters *m*, *S* and $E_{1}$ the model was not able to achieve a good fit (p≫0.05 for the ANOVA *F*-test) but it is still possible to extract information from the values of coefficients. The power index *m* reflects the underlying mechanism of the liquid penetration process and shows significant dependency on only porosity with $a_{1}=-25.00$. The apparent swelling rate *S* is poorly correlated to any of the factors, but its profile of coefficients, high in $a_{1}$ and $a_{3}$ resembles the response of *P*, implying the swelling rate is controlled by the liquid penetration rate which is intuitively straightforward to accept. The extent of swelling $E_{1}$ was found to be only significantly dependent upon SDI concentration at 90% confidence interval, with a negative coefficient $a_{2}=-5.914$, which implies that this initial swelling at 1 mm penetration depth is more likely attributed to the rapid volume expansion of MCC in stead of SDI upon absorbing water.

## 4. Conclusions

This study has demonstrated the ability of TPI to perform consistent quantitative analysis on water ingress and swelling simultaneously of tablets with complex formulations. The method captures the whole timespan of the process with high time resolution. The TPI data provide a good complementary mechanistic insight to the USP dissolution/disintegration tests. Together they disentangle the different contributions towards disintegration from water ingress, swelling and shear erosion.

The kinetics of water ingress for all samples follow a power-law fitting exceptionally well.

Adding NaCl is found to increase both liquid penetration rate and dissolution rate, though it seems to make little difference when the NaCl is increased from 5 to 10 wt%. It would be interesting in future work to investigate formulations with smaller step change in NaCl between 0 to 5 wt%.

A factorial analysis has revealed that compared to liquid penetration, the USP dissolution rate is more sensitive to SDI concentration and less significantly affected by NaCl concentration, while both processes remains dominated by porosity in the range of 10 to 20% porosity. The one-dimensional extent of swelling detected at 1 mm liquid penetration appears to reflect the contribution from MCC volume expansion.

Beyond demonstrating the ability of TPI to characterise fundamental rate processes within disintegrating and dissolving high complexity formulations, this study also provided valuable insight into the nature of NaCl as a disintegration aid in solid dispersion formulations. The results showed that NaCl significantly influences the water penetration rate, owing to the osmotic potential of the material and contributions to gelling inhibition. This supports the use of NaCl as an additive in solid dispersion tablet formulations to improve disintegration and dissolution.

## Figures and Tables

**Figure 1 pharmaceutics-13-00933-f001:**
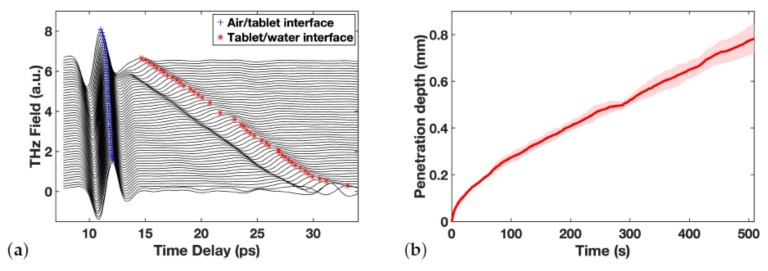
(**a**) An example of resulting TPI waveforms over time, with two interfaces tracked and marked. (**b**) Averaged penetration depth vs. time for a batch, errors illustrated as the shadowed area.

**Figure 2 pharmaceutics-13-00933-f002:**
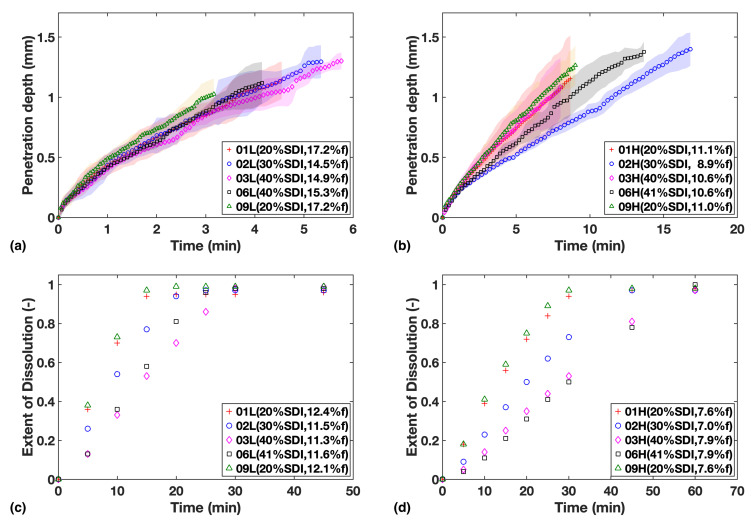
Study 1—Liquid penetration vs. time from TPI data for (**a**) Low compaction batches (**b**) High compaction batches, which is to be compared with USP dissolution profiles for (**c**) Low compaction batches (**d**) High compaction batches. Mass fraction of SDI and mean batch porosity labelled in the legends.

**Figure 3 pharmaceutics-13-00933-f003:**
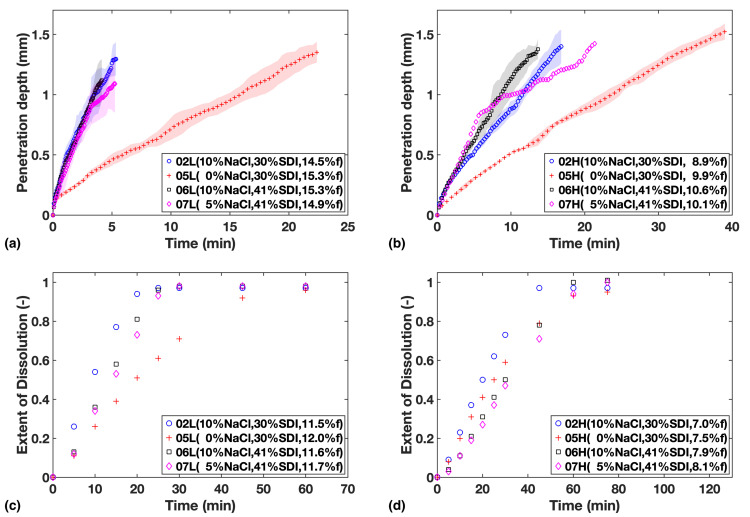
Study 2—Liquid penetration vs. time from TPI data for (**a**) Low compaction batches (**b**) High compaction batches, which is to be compared with USP dissolution profiles for (**c**) Low compaction batches (**d**) High compaction batches. Mass fraction of NaCl, SDI and mean batch porosity labelled in the legends.

**Figure 4 pharmaceutics-13-00933-f004:**
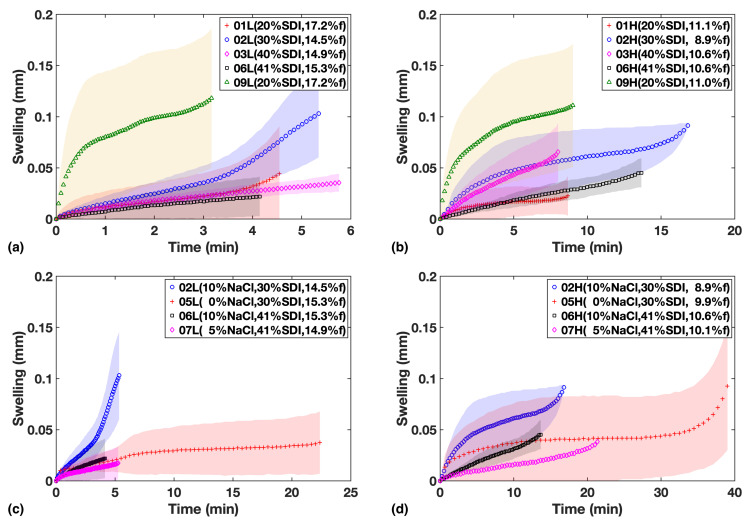
Swelling in mm vs. time for formulations in study 1 (**a**,**b**) and study 2 (**c**,**d**). Left column shows low compaction batches and right column shows High compaction batches.

**Figure 5 pharmaceutics-13-00933-f005:**
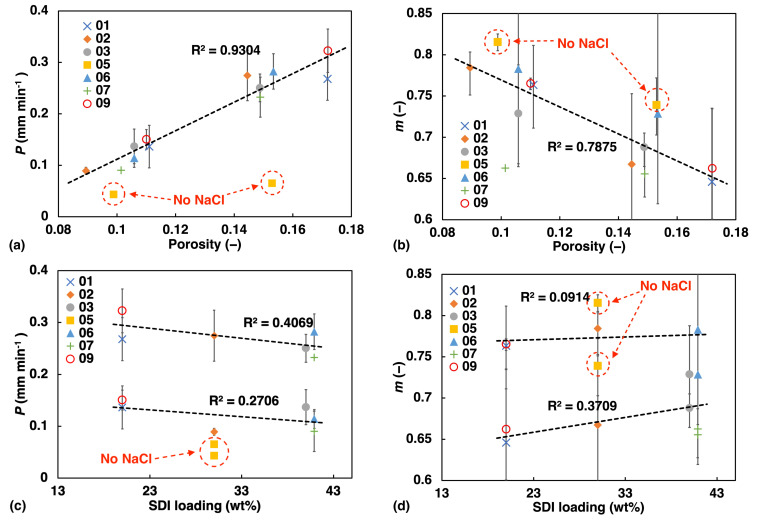
Liquid transport as quantified using the power-law fit of the TPI measurement data plotted against porosity (**a**,**b**); and against SDI loading (**c**,**d**).

**Figure 6 pharmaceutics-13-00933-f006:**
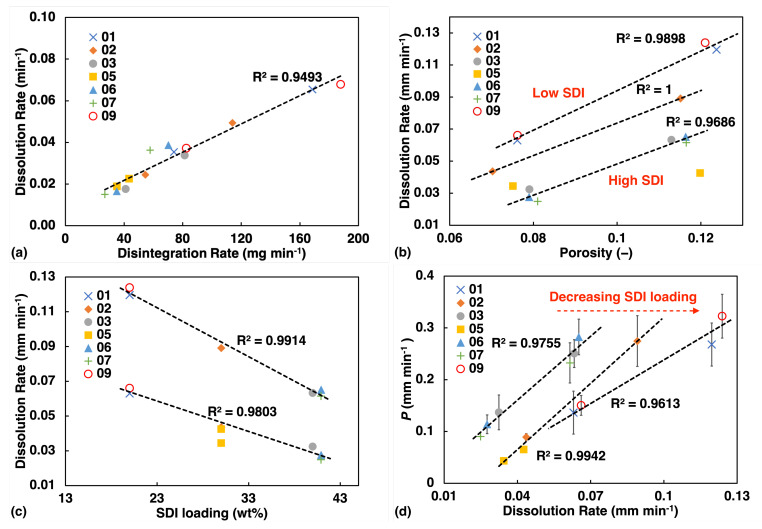
USP test results (**a**) Disintegration test showing high linearity with dissolution rate. (**b**) Dissolution test: rate constant from fitting the dissolution curve, plotted against porosity. (**c**) Dissolution rate plotted against SDI loading. (**d**) Correlation between the water ingress rate constant *k* and the dissolution rate from the tablets.

**Figure 7 pharmaceutics-13-00933-f007:**
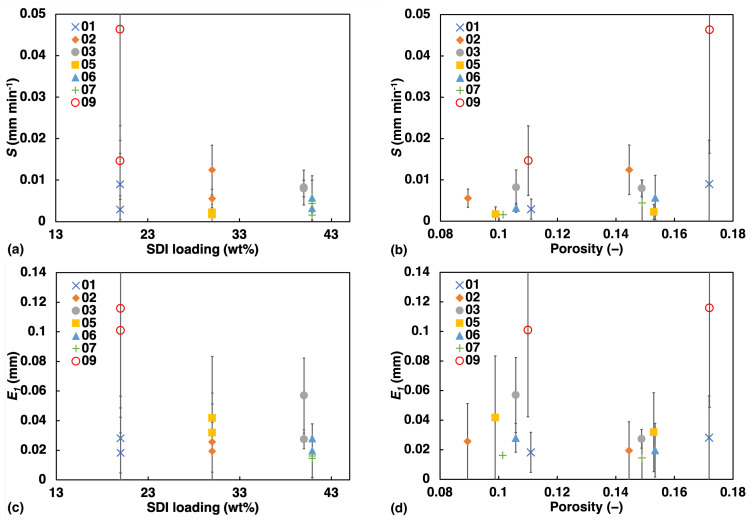
Apparent swelling rate *S* (**a**,**b**) and the extent of swelling $E_{1}$ (**c**,**d**) at 1 mm liquid penetration against: Left: SDI, Right: Porosity.

**Table 1 pharmaceutics-13-00933-t001:** Tablet formulations. In addition to the listed ingredients all formulations contain croscarmellose sodium (10 wt%), magnesium stearate (1 wt%) and sodium stearyl fumarate (1 wt%).

Key Ingredients (wt%)	F01	F02	F03	F05	F06	F07	F09
API1 (ruzasvir SDI)	20.00	30.00	40.00	30.00	40.91	40.90	20.00
API2 (uprifosbuvir)	15.00	15.00	15.00	15.00	20.45	20.50	10.00
Sodium chloride	10.00	10.00	10.00	0.00	10.00	5.00	10.00
Mannitol	14.23	10.85	7.46	14.18	5.34	7.00	15.90
Microcrystalline cellulose	28.46	21.69	14.92	28.36	10.67	14.00	31.80
Colloidal silicon dioxide	0.307	0.460	0.610	0.460	0.627	0.600	0.310
Study 1: SDI	X	X	X		X		X
Study 2: NaCl		X		X	X	X	

**Table 2 pharmaceutics-13-00933-t002:** Summary of the true density and calculated porosity (Equation (Equation 1)) for all batches.

Formulation	True Density	Porosity	Porosity
	(g mL^−1^)	Low Compaction (-)	High Compaction (-)
F01	1.453	0.1718 ± 0.0013	0.1110 ± 0.0033
F02	1.414	0.1445 ± 0.0021	0.0894 ± 0.0043
F03	1.378	0.1488 ± 0.0041	0.1059 ± 0.0021
F05	1.378	0.1529 ± 0.0018	0.0988 ± 0.0026
F06	1.364	0.1534 ± 0.0027	0.1059 ± 0.0076
F07	1.347	0.1489 ± 0.0049	0.1014 ± 0.0026
F09	1.463	0.1720 ± 0.0010	0.1100 ± 0.0053

**Table 3 pharmaceutics-13-00933-t003:** Coefficients from multiple regression analysis. Non-significant coefficients at 95% confidence interval (p>0.05) are shaded. The coefficients accepted at 90% confidence interval (0.10>p>0.05) are in lighter shade.

*Y*	$D/\sigma_{D}$	*p*-Value	$P/\sigma_{P}$	*p*-Value	$m/\sigma_{m}$	*p*-Value	$S/\sigma_{S}$	*p*-Value	$E/\sigma_{E}$	*p*-Value
$a_{1}$ (Porosity)	30.24	<0.0001	26.52	<0.0001	−25.00	0.0077	13.16	0.1446	−3.397	0.7150
$a_{2}$ (SDI)	−6.220	<0.0001	1.065	0.4076	−2.312	0.3656	−3.999	0.1693	−5.914	0.0716
$a_{3}$ (NaCl)	9.207	0.0026	13.51	0.0009	−4.021	0.4972	7.305	0.2742	5.525	0.4396
$R^{2}$	0.9248		0.8839		0.5446		0.4426		0.3411	
ANOVA *p*-value	<0.0001		<0.0001		0.0417		0.1063		0.2247	

## Data Availability

All datasets generated and/or analysed during this study are available from the corresponding author on reasonable request. Datasetlicense-CC-BY 4.0. Dataset: https://doi.org/10.17863/CAM.71617 (acessed on 1 May 2021).

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
