# Peer review of "Insights into the Control of Drug Release from Complex Immediate Release Formulations"

_pharmaceutics, 2021, doi:10.3390/pharmaceutics13070933_

Round 1
Reviewer 1 Report
TPI and dissolution tests are performed to investigate the rate of water penetration, the extent of swelling, and USP dissolution rate for tablets for which porosity, SDI content, and NaCl are varied.
The TPI experiments and results are extremely interesting but the connection with dissolution experiments is not so clear or so straightforward as claimed by the authors.
There are some major points that should be addressed
1) No error bars are reported for dissolution data and are extremely important for USP data
2) Dissolution rate for 05H (no NaCl, 30% SDI) is higher than 06H and 07H. Please comment on this because it seems to be in contradiction with your analysis lines 202-210.
3) Can you comment on the significant differences between swelling behaviors of 09L and 01L in figure 4A and trace the connection with dissolution data in figure 2a ? I see some data contradicting the very general sentence lines 229-232.
4) Sentence 217-223 should be supported by literature references.
5) Is the best fit range of k and m parameters limited to 0-1 mm of penetration depth? I think so although the R2 value cannot be so high for data like 07H. Moreover P is affected by m, given its definition, eq. 3. therefore it makes no sense to investigate both P and m, P would be sufficient. Maybe the simple t_(1mm) time to reach 1 mm depth would be already a representative quantity.
6) Show a picture of at least one swelling curve, its best fit with the polynomial function adopted, and highlight the two inflection points. It's quite hard to recognize an S shape for many of the data shown in figure 4.
7) Factorial analysis limited to the linear terms (eq. 5) is not capable to show the joint effects of two parameters, e.g. porosity and SDI concentration. Quadratic terms must be included.
8) Fugures 7 A-d are not clear. Response surfaces would better show the influence of different independent variables
9) line 143. The temperature of water used was 20^oC. Please comment on this choice.
Suggestion:
It would be more interesting to show the correlations between dependent variables D,P, m and S through PCA (principal component analysis)
minor points
error bars in figure 3 (b) are missing for 07H data.
error bars in figure 4 (c) are missing for 07L data
error bars in figure 4 (d) are missing for 07H data.
lines 187-190 what is a gradient of dissolution profile? a gradient is a vector quantity. The rate of growth would be more appropriate. Please reformulate the sentence.
line 244 "firstmm" should be "first 1 mm" ?
Author Response
Please see our responses in the attached document.

Reviewer 2 Report
The article is written and presented well. There are some comments that should be addressed. (Highlight changes in revised submission).
- The authors should comment on the omission of formulations F4 and F8 from Table 1 and Table 2.
- More information on the composition of the FDC tablet is required including the amounts of the SDI API and excipients.
- Was the porosity of the prepared tablets confirmed using a porosimeter? If so, please can the authors include this as well. If not, please include further details and references for the calculation method used.
- For the TPI, the authors should state why the temperature used was 20ºC and not physiological temperature.
- The authors should also state which temperature was used for the USP dissolution and disintegration tests as well as the equipment that was used. Also, was there any media extraction and replacement, and was any analysis of the extracted media undertaken and what are the details of these?
- The standard deviations of the individual tests in Figures 2-4 should be included.
Author Response
Please see our attached response.

Reviewer 3 Report
The manuscript leads with the ability of terahertz pulsed imaging as a non-destructive technique for exploring the inter-molecular structure of tablet and the transport water dynamics from dissolution medium into the porous tablets, and simultaneously to perform consistent quantitative analysis of swelling rate, disintegration time, drug dissolution and release profile.
The approach is interesting, and the paper has the potential to contribute to technical and scientific advances. However, minor and major changes are necessary.
Minor changes
- When you use an abbreviation in both the abstract and the text, define it in both places upon first use.
- Remove the per cent sign (0%)
- The text sequence was interrupted by Tables 1 and 2. Adjust the position of the Tables in the text.
- Make the sentence readable for a better understanding of the aims (Lines 96-99).
- What does the mark ( - ) in Table 2 mean?
- Quote the mass of the tablets (line 109).
Major Changes
- Solid dispersion has a clear definition. Make clear what is solid dispersion intermediate in the context of the manuscript.
- The authors explore suitably the factorial analysis and show a good discussion of results. However, the phenomenological discussion of the main results is missed. The phenomena observation are required to make better the scientific approach.
- The statement (line 372) is out of context.
Author Response
Please see our attached response.
